# Engineering *Escherichia coli*-Derived Nanoparticles for Vaccine Development

**DOI:** 10.3390/vaccines12111287

**Published:** 2024-11-18

**Authors:** Shubing Tang, Chen Zhao, Xianchao Zhu

**Affiliations:** 1Shanghai Reinovax Biologics Co., Ltd., Pudong New District, Shanghai 200135, China; jzhu@reinovax.com; 2Shanghai Public Health Clinical Center, Fudan University, Shanghai 201058, China

**Keywords:** subunit vaccines, molecular and protein engineering, reassembly, virus-like particles, *E. coli*-derived nanovaccines

## Abstract

The development of effective vaccines necessitates a delicate balance between maximizing immunogenicity and minimizing safety concerns. Subunit vaccines, while generally considered safe, often fail to elicit robust and durable immune responses. Nanotechnology presents a promising approach to address this dilemma, enabling subunit antigens to mimic critical aspects of native pathogens, such as nanoscale dimensions, geometry, and highly repetitive antigen display. Various expression systems, including *Escherichia coli* (*E. coli*), yeast, baculovirus/insect cells, and Chinese hamster ovary (CHO) cells, have been explored for the production of nanoparticle vaccines. Among these, *E. coli* stands out due to its cost-effectiveness, scalability, rapid production cycle, and high yields. However, the *E. coli* manufacturing platform faces challenges related to its unfavorable redox environment for disulfide bond formation, lack of post-translational modifications, and difficulties in achieving proper protein folding. This review focuses on molecular and protein engineering strategies to enhance protein solubility in *E. coli* and facilitate the in vitro reassembly of virus-like particles (VLPs). We also discuss approaches for antigen display on nanocarrier surfaces and methods to stabilize these carriers. These bioengineering approaches, in combination with advanced nanocarrier design, hold significant potential for developing highly effective and affordable *E. coli*-derived nanovaccines, paving the way for improved protection against a wide range of infectious diseases.

## 1. Introduction

Vaccines represent one of the most cost-effective and impactful strategies for controlling infectious diseases. Vaccination played a critical role in mitigating the global spread of severe acute respiratory syndrome coronavirus 2 (SARS-CoV-2), saving millions of lives [1]. Subunit vaccines have garnered significant attention for their favorable safety profile, with a substantial proportion of clinical trials dedicated to their development during the SARS-CoV-2 pandemic [2]. However, subunit vaccines possess inherent limitations, including suboptimal immunogenicity and an insufficient capacity to elicit cellular immune responses [3]. Nanotechnology has emerged as a valuable tool to address these challenges, enabling the display of subunit protein antigens on particle surfaces in a highly ordered, conformationally authentic, and repetitive array [4]. These nanoparticle vaccines enable strengthening immune responses without compromising safety by mimicking the nanoscale size, geometry, and multivalent antigen presentation of native pathogens [5].

A variety of expression systems, from *E. coli* to mammalian cells, are employed for the production of protein-based nanoparticle vaccines [6]. *E. coli* is widely utilized as a nanoparticle expression system due to its affordability, high yield, and rapid production cycle [7]. However, proteins expressed in *E. coli* can exhibit challenges, including improper disulfide bond formation, poor solubility, an inability to self-assemble into nanoparticle architectures, lack of post-translational glycosylation modifications, and endotoxin contamination [8,9]. Endotoxins can induce fever reactions and even heat shock in mammals; however, these contaminants can be effectively minimized during protein purification processes [10]. The strategies employed to facilitate proper folding and self-assembly of proteins into nanoparticles within the *E. coli* expression system will be further discussed below.

The success of *E. coli*-derived virus-like particle (VLP) and nanoparticle vaccines is evident in both licensed products and ongoing clinical trials, demonstrating the platform’s potential for addressing global health challenges (Table 1). One notable example is the covalent linkage of the receptor-binding domain (RBD) of SARS-CoV-2 onto nanoparticles expressed by *E. coli*, LYB001, which has entered clinical phase III (NCT05664932). The first licensed *E. coli*-derived VLP vaccine, Hecolin^®^, was launched in China in 2012 [11,12]. An N-terminal and C-terminal truncation of the hepatitis E virus (HEV) capsid protein (amino acids 368-606) resulted in the HEV239 protein, which was highly expressed in *E. coli* as inclusion bodies. These inclusion bodies were solubilized using 4 M urea with high salt to promote nanoparticle assembly and subsequently purified through two polishing chromatography steps, achieving a purity of 95–98% [13]. Another licensed *E. coli*-derived VLP vaccine, Cecolin^®^, is a bivalent human papillomavirus (HPV) vaccine [14]. The L1 proteins of HPV16 and HPV18 were N-terminally truncated, yielding soluble proteins that were sequentially purified using SP Sepharose and CHT II resin. Following the removal of reductants introduced in previous steps, VLPs were reassembled through a dialysis process [15]. Additionally, *E. coli* has been successfully employed for the production of norovirus VLPs. The VP1 protein, the major capsid protein of norovirus, was initially insoluble *in E. coli* but achieved high solubility with the help of host tRNA. A low-pH dialysis procedure was then employed to induce self-assembly of VP1 proteins into VLPs [16]. Through the combined application of molecular design and protein engineering strategies, the potential of *E. coli* in VLPs manufacturing has been significantly expanded.

Beyond their direct application as vaccines, VLPs and protein-based nanoparticles have found widespread use as vaccine platforms for the delivery of desired antigens [21]. Commonly employed nanocarriers include VLPs formed from hepatitis B virus core antigen (HBc) [22], bacteriophage capsid protein AP205 and Qβ [23], and nanoparticles of ferritin [24], Lumazine Synthase (LS) [25], encapsulin [26], computationally designed mI3 based from 2-keto-3-deoxy-6-phosphogluconate (KDPG) [27], S domain [28] and P domain [29] of norovirus VP1 (Figure 1). Despite their significant potential for enhancing immunogenicity, nanoparticle delivery strategies often encounter challenges. Loaded antigens can create steric hindrance, potentially leading to nanoparticle disassembly, and improper antigen orientation can result in suboptimal immunogenicity [21]. The subsequent sections will delve into the lessons learned and engineering technologies employed to address these challenges and optimize the rational display of antigens onto nanocarriers, maximizing vaccine efficacy.

## 2. Immunogenicity Potentiating Mechanism by Nanoparticles

Effective immune responses rely heavily on secondary lymphoid tissues, particularly the T cells, B cells, and antigen-presenting cells (APCs) residing within lymph nodes [30]. The activation levels of T cells and B cells are closely correlated with antigen accumulation in these lymph nodes [31]. Therefore, targeted delivery of antigens to lymph nodes is crucial for optimal immune responses. Proteins or particles smaller than 10 nm readily diffuse into blood vessels and are rapidly cleared, while particles larger than 200 nm can be engulfed and presented by APCs [32,33]. Larger particles, such as aluminum adjuvant particles adsorbed with antigens, recruit immune cells to the injection site. These cells, such as dendritic cells (DCs), engulf the antigen-adjuvant complex and mature, eventually migrating back to the lymph nodes through a process called chemokine homing. This ‘reverse targeted’ cell infiltration phenomenon effectively delivers the antigen to the lymph nodes, where it can be presented to T cells and B cells [34]. In mouse and rat models, nanoparticles with a size range of 20 to 100 nm have been shown to efficiently enter and reside within lymph nodes (Figure 2) [35,36,37]. This efficient lymphatic drainage of nanoparticles may be a passive transport process, as transport efficiency remained largely unaffected by the absence of migrating dendritic cells (DCs) in CCR7^−/−^ mice [38].

Compared to protein antigens, nanoparticle-based vaccines offer several advantages, including efficient targeting lymph nodes, increasing the likelihood of interaction with APCs, and facilitating antigen internalization and presentation efficiency by APCs [39,40]. The highly repetitive array of antigens on the nanoparticle surface enables high-affinity binding to IgM, subsequently recruiting complement component 1q (C1q) and activating the classical complement cascade. In addition to IgM, nanoparticles also bind to pentaxins, further enhancing antigen internalization through interactions with Fc receptors [41]. Following internalization by APCs, antigens are degraded into peptides within the endosome-lysosome compartment. These peptides then form complexes with MHC class II, upregulate the expression of the co-stimulatory factor CD40 on the surface of APC cells, and ultimately activate CD4+ T helper cells via the CD40-CD40L signaling pathway [42]. Moreover, nanoparticles promote peptide cross-presentation through the MHC class I pathway, leading to elevated expression levels of the co-stimulatory factors CD80/CD86 and subsequent activation of CD8+ T-cell responses through the CD80/CD86-CD28 signaling pathway [43,44,45]. This, in turn, triggers cytotoxic T lymphocyte (CTL)-mediated cellular immune responses to eliminate intracellular pathogens [46].

Following their passive drainage to lymph nodes via lymphatic vessels, nanoparticles can also diffuse to the B cell region and directly interact with follicular B cells, contributing to the induction of robust antibody responses. The highly repetitive and well-ordered arrangement of antigens on the nanoparticle surface resembles pathogen-associated molecular patterns (PAMPs), facilitating cross-linking of B cell receptors (BCRs) [5,47]. The display of 15–20 antigenic molecules at a distance of 5–10 nm has been shown to effectively activate B cells [48]. Moreover, the repetitive display of antigens enhances complement activation through the CD19-CD21 complex, further contributing to B cell activation. Simultaneous cross-linking of BCRs and CD19-CD21 lowers the threshold of BCR activation required for B cell activation [49,50]. Even in the absence of adjuvants, low concentrations of nanoparticles are sufficient to induce durable and robust antibody immune responses [51].

## 3. Development of *E. coli*-Derived Nanovaccines

### 3.1. Development of E. coli-Derived HEV Vaccines

Hepatitis E virus (HEV), a single-stranded positive-sense RNA virus, poses a significant health burden, particularly in developing countries [52]. The HEV genome encodes three open reading frames (ORFs), with ORF2 responsible for the capsid protein that self-assembles into 180-mer icosahedral structures (T = 3) [53]. The N-terminal 110 amino acids (aa) of the capsid protein are rich in arginine, facilitating interactions with RNA. The C-terminal region comprises three functional domains: the S domain (aa 118–313), the P1 domain (aa 314–453), and the P2 domain (aa 454–606) [54,55]. In 1997, Tiancheng Li et al. expressed an N-terminal truncation of the HEV capsid protein (aa 112–660) using a baculovirus/insect cell (Tn5 cell) expression system, purifying the product from the supernatant of the culture medium. They observed the formation of VLPs with a diameter of 23.7 nm, slightly smaller than native virions (27 nm) [56]. In 2011, the same group revealed that an N-terminal deletion of 100 aa from the capsid protein of rat HEV did not disrupt VLP structure. Interestingly, they identified two distinct VLP sizes with diameters of 24 nm and 35 nm, respectively [57]. These studies suggest that the baculovirus expression system allows for deletions within the capsid proteins of both human and rat HEV without compromising their assembly into viral particles.

In 2001, Jizong Zhang et al. expressed a truncated human HEV capsid protein (aa 394–604) in *E. coli*. They discovered that this protein formed conformational dimers that were recognized by serum from volunteers whose serum reacted with HEV [58]. While this dimeric protein exhibited protective properties, it failed to induce sufficient immune responses. In 2005, Shaowei Li et al. constructed a series of N-terminal and C-terminal deletions, along with various mutations, within the HEV capsid protein. Their findings revealed that six hydrophobic residues (Ala597, Val598, Ala599, Leu601, and Ala602) were critical for dimer formation. Importantly, a specific truncation (aa 368–606) termed p239 was self-assembled into 23 nm particles [59]. Subsequent research further elucidated the factors contributing to the self-assembly of p239 homodimers into nanoparticles. In 2013, Chunyan Yang et al. demonstrated that salt treatment facilitated the assembly of p239 homodimers into nanoparticles in 4 M urea. They tested different concentrations of ammonium sulfate, sodium sulfate, sodium chloride, or ammonium chloride, finding that sodium sulfate was the most effective. Direct refolding of p239 from 4 M urea resulted in incomplete nanoparticle reassembly with two distinct sizes, while homogeneous nanoparticles were obtained with the assistance of ammonium sulfate. Furthermore, site-directed mutagenesis targeting the hydrophobic core between aa 368–394 revealed that hydrophobic residues Leu372, Leu375, and Leu395 were crucial for particle assembly [60]. These investigations shed light on the reassembly mechanism of p239 nanoparticles, leading to the development of Hecolin^®^ and providing valuable insights for addressing the challenges of obtaining native nanoparticles from inclusion bodies.

Building upon the accumulated knowledge regarding the self-assembly of p239 into nanoparticles, Xiao Zhang et al. developed the hydrophobic region (aa 369–460) of p239 as a universal nano-delivery vector. A variety of antigens, including HA1 of influenza, gp41/gp120/p24 of human immunodeficiency virus (HIV), HBsAg, and L2 of HPV16, were fused to the C-terminus of this nanocarrier. All recombinant proteins formed inclusion bodies that were solubilized using 4 M urea. Subsequent ion-exchange and hydrophobic interaction chromatography were employed to purify the products, followed by a refolding process to harvest nanoparticles. These nanoparticles elicited antigen-specific IgG titers that were 100-fold higher than those induced by their non-assembling protein counterparts [61].

### 3.2. Development of E. coli-Derived HPV Vaccines

HPV is a double-stranded DNA virus encased in a non-enveloped capsid composed of 72 pentameric L1 and a controversial number of L2 monomers [62,63]. HPV infection is a major cause of cervical cancer, accounting for approximately 5% of all cancers. Over 200 HPV genotypes have been identified, with more than 18 posing a significant risk for cervical cancer [64]. In 2006, *Saccharomyces cerevisiae* was employed to produce HPV VLPs formed by the major capsid protein L1 as prophylactic vaccines. Unlike HPV18, the L1 proteins of HPV6, 11, and 16 assembled into heterogeneous VLPs with diameters ranging from 30 nm to 50 nm, rather than the 60-nm homogenous VLPs typically observed. An iterative process was conducted to screen for optimal buffer conditions for disassembly. After disassembly and reassembly, uniform VLPs with a diameter of 60 nm were obtained, and the stability of HPV VLPs was significantly enhanced [65]. The use of the reductant DTT was based on a previous study demonstrating that three cysteine residues within L1, C175, C185, and C428 are essential for intermolecular bonding and VLP assembly [66]. This disassembly/reassembly treatment not only yielded more homogeneous VLPs but also induced a higher proportion of conformationally and neutralizing antibodies [67].

In addition to disassembly/reassembly treatments, truncation strategies have also been explored to improve the assembly of HPV VLPs. In 2006, Arvind Varsani et al. conducted N-terminal and C-terminal truncations of the HPV16 L1 protein within a baculovirus/insect cell (Sf21) expression system, revealing that VLP structure was affected by deletions [68]. Another study in the same year deleted the first aa 43 of HPV16 L1 and fused it with a GST tag at its N-terminus. This resulted in the formation of heterogeneous 30–40 nm nanoparticles in *E. coli* when recombinant proteins were induced [69]. To generate homogeneous HPV16 and HPV18 VLPs in *E. coli*, Ying Gu et al. established a series of N-terminal deletions of L1 proteins and performed disassembly/reassembly treatment. Both HPV16 and HPV18 L1 proteins were highly expressed when the first four residues were deleted, and the manufacturing procedure was scaled up to 500 L. The soluble, truncated proteins were sequentially purified using SP Sepharose and CHT II resin chromatography in the presence of DTT as a reductant. Uniform VLPs were obtained following a reassembly process, eliciting a comparable immune response to the licensed GARDASIL^®^ 2 vaccines [15]. This N-terminal truncation strategy proved not only suitable for bivalent HPV vaccine (HPV16 and HPV18) production but also adaptable for generating nine-valent HPV vaccines (HPV6, HPV16, HPV18, HPV31, HPV33, HPV45, HPV52, HPV58) in the *E. coli* system. Kaihang Wang et al. utilized CRISPR-Cas9 genome editing technology to inactivate eight genes (lpxL, lpxM, lpxP, eptA, pagP, kdsD, msbA, and gutQ) related to LPS synthesis in the *E. coli* ER2566 strain, reducing LPS synthesis during plasmid-based L1 protein expression. To get rid of the use of antibiotics, they further developed a plasmid-free system by inserting the L1 gene of HPV into the aforementioned genes and discovered that six genes (lpxM, lpxP, lpxL, eptA, gutQ, and kdsD) were suitable for insertion. They constructed a multicopy expression cassette that produced nine-valent HPV VLPs with a yield 1.5–2 times higher than the plasmid expression system while reducing endotoxin to very low levels [70].

To date, over 200 HPV genotypes have been identified, with more than 18 classified as highly pathogenic and linked to the development of cancers [71]. While L1 VLPs have shown efficacy in providing genotype-specific protection against HPV, cross-protection has been limited [72]. The demand for the development of higher-valent HPV vaccines is therefore critical. A promising strategy for developing multivalent HPV vaccines involves the design of chimeric VLPs through loop swapping. Based on the structure of HPV L1, five conserved surface loops (BC, DE, EF, FG, and HI) are considered the primary determinants of HPV genotype specificity [73,74]. A trivalent HPV33/58/52 VLP vaccine was created by replacing the HPV58 VLP scaffold with the HPV33 BC loop and the HPV52 HI loop. This chimeric VLP elicited high neutralizing antibody titers against all three genotypes. Similarly, four other trivalent HPV VLPs (HPV16/35/31, HPV56/66/53, HPV39/68/70, and HPV18/45/59) were generated using this approach [75]. This loop-swapping strategy holds significant potential for the development of higher-valent HPV vaccines.

Two cysteine residues, C175 and C428, form disulfide bonds that are essential for the assembly of HPV VLPs [76]. A novel approach to HPV VLP reassembly was developed to generate hybrid VLPs. L1 proteins carrying single C175A or C428A mutations were capable of forming pentamers but lost the ability to assemble into VLPs. By combining distinct pairs of C175A and C428A mutants from nine HPV genotypes in a defined stoichiometry, nine-valent hybrid HPV VLPs were generated. This new hybrid VLP vaccine induced comparable neutralizing antibody titers to those elicited by the Gardasil 9 vaccine [77].

### 3.3. Development of E. coli-Derived Norovirus Vaccines

Norovirus is a leading cause of viral acute gastroenteritis, primarily affecting the elderly, children, and immunocompromised populations worldwide [78]. The most advanced norovirus vaccine, HIL-214, is currently in phase III clinical trials (NCT05836012, NCT05281094). HIL-214 contains VP1 VLPs of genotypes GI.1 and GII.4, expressed using a baculovirus/insect cell (Sf9) expression system. When genotype GI.1 VP1 was expressed in this system, it assembled into two sizes of VLPs with diameters of 23 nm (T = 1) and 38 nm (T = 3). Importantly, these VLPs tolerated N-terminal, C-terminal, and intermediate truncations [79,80]. Using the same cell lines, GII.4 VP1 also formed two sizes of VLPs, and N-terminal truncation led to the generation of smaller VLPs with an average size of 21 nm [81]. The full-length norovirus VP1 protein formed inclusion bodies when expressed in *E. coli*. Inspired by these studies, Yuqi Huo et al. constructed two truncations of genotype GII.4 VP1 (N26, N38) and expressed these truncated proteins using the cold shock expression vector pCold in *E. coli*. Although the truncated proteins assembled into VLPs, their yields were relatively low. Interestingly, two protein bands were observed for these truncated mutants indicating that proteins degraded during manufacture process [82]. Another intriguing approach to improve norovirus VP1 solubility in *E. coli* involves utilizing host tRNA as a molecular chaperone. A 6xHis tag, a tRNA interacting domain (tRID), and a TEV cleavage motif were introduced to the N-terminus of GII.4 VP1. The soluble recombinant tRID-VP1 existed as a dimer after nickel chromatography purification. tRID was cleaved from recombinant protein following TEV digestion, along with which host tRNA was removed. A low-pH dialysis process was then performed for the assembly of VLPs, resulting in uniform 40 nm VLPs [16].

### 3.4. HBc VLPs as a Nanocarrier for Antigen Display

Beyond their direct use as vaccines, VLPs and nanoparticles have been extensively explored as nanocarriers for antigen delivery (Table 2). The 180-mer HBc (T = 3) VLP is among the most widely utilized nanocarriers [83]. The N-terminus of HBc could accommodate a single copy of the influenza virus M2e epitope, while tandem insertion of two or three copies of the M2e epitopes leads to the formation of inclusion bodies. Above inclusion bodies were solubilized using 4 M urea, and VLPs were obtained through a refolding process [84,85]. In addition to its N-terminus, the major antigenic dominant region (MIR) and the C-terminus of HBc could also tolerate the insertion of exogenous antigens (Figure 3A) [86,87]. When antigens were introduced, recombinant proteins might form inclusion bodies, requiring a refolding-reassembly treatment to obtain VLPs [85,88]. Proper display of antigens onto HBc VLPs often involves trial and error. For example, inserting the full-length or N-terminal domain of Factor H binding protein (FHbp) from serogroup B meningococci (MenB) variant 1 into the HBc C-terminus disrupted VLP formation. However, incorporating the C-terminal domain of variant 1 FHbp (CFHbp) into the HBc C-terminus did not compromise VLP self-assembly. Despite successful incorporation, CFHbp failed to induce protective antibodies, likely due to improper conformation. Surprisingly, inserting the naturally trimeric NadA of MenB into the MIR did not disrupt VLP assembly. Electron density analysis indicated that NadA partially folded into a trimer, and immune studies demonstrated that the recombinant VLPs were sufficient to induce protective antibodies [89,90].

There are two key challenges associated with inserting exogenous antigens into VLPs: maintaining VLP assembly and preserving the authentic tertiary structure of the incorporated antigens. One strategy to mitigate the steric hindrance caused by inserted antigens involved engineering two consecutive HBc units as a tandem core. This approach allowed for a single antigen, rather than two, to be inserted into the MIR, providing more space for the antigen to adopt a proper conformation (Figure 3B) [91]. Another method to stabilize HBc VLPs involved introducing the D29C and R129C mutations, which induced an artificial disulfide bond to strengthen the dimer-to-dimer interface. Freeze-thaw cycle analysis demonstrated a significant improvement in HBc VLP stability after the introduction of this disulfide network [92].

An alternative approach to overcome the challenges of antigen insertion involved independently generating antigens and VLPs, subsequently coupling them via chemoenzymatic reactions. In 2016, a modular platform utilizing sortase-mediated site-specific tagging was devised using HBc VLPs [93]. HBc was split into two parts at the MIR site (N-core with amino acids 1–79 and C-core with amino acids 80–185), and an LPETGG motif was fused to the C-terminus of the N-core, making it accessible for covalent conjugation of antigens with oligoglycine at their N-terminus using sortase (Figure 4A) [94]. While this method demonstrated high efficiency in coupling a protein antigen and two peptides onto the HBc nanocarrier, it required a large excess of oligoglycine reactants due to the reversibility of the reaction [95]. Therefore, split intein-mediated *trans*-splicing was implemented to couple antigens onto the HBc nanocarrier, enabling an ultrafast reaction with minimal excess substrates [96]. The gb1 tag, the B1 domain of streptococcal protein G, was incorporated to enhance the solubility and yield of reactants, which could be removed through intein-mediated *trans*-splicing (Figure 4B). Green fluorescent protein (GFP) and the nucleocapsid protein (NP) of SARS-CoV-2 were successfully and efficiently coupled onto the HBc nanocarrier using this method. Compared to the aforementioned proteins, nanoparticles induced 10-fold higher antigen-specific IgG titers [97]. Furthermore, SpyTag/SpyCatcher technology has also been employed to display antigens onto HBc nanocarriers (Figure 4C) [98].

**Table 2 vaccines-12-01287-t002:** Genetic fusion of desired molecules to different sites of nanocarriers.

Nanocarrier	Molecules	Insertion Site	References
HBc	Influenza M2e	N-terminus	[84]
	One to three copies of Influenza M2e	N-terminus, MIR	[85]
	HBV preS (20–47), HCV core (1-60)	MIR, C-terminus	[86]
	FHbp, NadA from serogroup B meningococci	MIR, C-terminus	[90]
Bacteriophage Qβ	6xHis	C-terminus	[99]
Bacteriophage AP205	SpyCatcher, SpyTag	N-terminus, C-terminus	[100,101,102]
Norovirus GII.4 VLPs	Influenza M2e, SpyTag	C-terminus	[103,104]
Ferritin	Influenza HA	N-terminus	[105]
	tumor-targeting proapoptotic peptide, GFP	N-terminus, C-terminus	[106]
Lumazine synthase	SARS-CoV-2 RBD	N-terminus	[107]
	gp120 outer domain immunogen of HIV	C-terminus	[108]
Encapsulin	6xHis, SP94 peptide	C-terminus, loop 42/43, loop 138/139	[109]
mI3	GFP	N-terminus, C-terminus	[110]

### 3.5. Bacteriophage AP205 Capsid VLPs as a Nanocarrier for Antigen Display

The capsid protein of bacteriophage AP205 spontaneously assembles into 180 copies (T = 3) of icosahedral VLPs, which readily accommodate the insertion of cargos at both the N-terminus and C-terminus, making it a promising nanocarrier platform [111,112]. In 2016, AP205 VLPs were engineered as a plug-and-display platform utilizing SpyTag/SpyCatcher technology for antigen display. SpyCatcher was fused to the N-terminus of the AP205 capsid protein, enabling covalent coupling of antigens containing SpyTag. Two malaria antigens, CIDR and Pfs25, were successfully displayed on this platform, and a single injection elicited strong immune responses [100]. This platform offers several key advantages. First, antigens can be produced using eukaryotic expression systems, eliminating concerns regarding post-translational modifications. Second, potential side effects arising from antigen-induced capsid misassembly are mitigated [113]. Homotypic hemagglutinin (HA) trimers from influenza A and heterotypic HA trimers from eight strains were covalently conjugated to AP205 VLPs using SpyTag/SpyCatcher, resulting in the generation of homotypic and mosaic nanoparticles. Both nanoparticle types elicited strong IgG responses; however, mosaic nanoparticles did not demonstrate broader immune protection than homotypic nanoparticles [101].

### 3.6. Bacteriophage Qβ Capsid VLPs as a Nanocarrier for Antigen Display

The capsid of bacteriophage Qβ forms dimers that subsequently assemble into 180-mer (T = 3) VLP [114]. CpG, when encapsulated within the inner cavity of bacteriophage Qβ VLPs, has been shown to induce robust T-cell responses, inhibiting the growth of mouse A20 lymphoma and improving survival rates [115]. While exogenous peptides or proteins can be inserted into the C-terminus of the bacteriophage Qβ capsid protein, VLP assembly is often compromised. For example, the genetic insertion of six consecutive histidines (6xHis) disrupts the formation of intact Qβ-derived VLPs. However, controlled mixing of wild-type Qβ capsid protein with Qβ capsid protein genetically fused with 6xHis results in hybrid VLPs that could be purified using nickel chromatography [99]. Human epidermal growth factor (EGF) was successfully incorporated into Qβ hybrid VLPs at a ratio of 5–12 copies per VLP particle. Additionally, EGF was coupled to the surface of Qβ VLPs using the azide-alkyne cycloaddition reaction (CuAAC). Both nanoparticle types facilitated the self-phosphorylation of EGF receptors, subsequently triggering apoptosis in A431 cells [116].

### 3.7. Ferritin Nanoparticle as a Vaccine Nanocarrier

Ferritin, ubiquitously found in nature, spontaneously assembles into octahedral nanocages composed of 24 monomeric units, serving as a crucial iron storage mechanism. These nanocages have been widely explored as carriers for drugs and vaccines [117]. While ferritins from both prokaryotic and eukaryotic organisms can be utilized as carriers, ferritin from *Helicobacter pylori* (*H. pylori*) is the most commonly employed [21]. Displaying eight HA trimers on the surface of ferritin nanoparticles had been shown to induce 10-fold higher hemagglutination inhibition antibody titers compared to the licensed inactivated vaccine [105]. Co-transfection of Expi293F cells with multiple plasmids containing the receptor-binding domain (RBD) of HA fused to the N-terminus of ferritin resulted in the generation of mosaic RBD-nanoparticles. These mosaic nanoparticles elicited broader antibody responses than mixtures of homotypic RBD-nanoparticles [118]. Additionally, ferritin had been engineered as a double-chambered nanocarrier by deleting its fifth α-helix. This engineered ferritin allowed for the introduction of a tumor-targeting proapoptotic peptide at its N-terminus while accommodating the insertion of GFP at its C-terminus, enabling dual functionalities [106].

Ferritin nanoparticles have emerged as a promising platform for antigen delivery and adjuvant co-delivery. However, our previous experience showed that genetic fusion of certain proteins, including rhizavidin or the Toll-like receptor 5 agonist CBLB502, with ferritin resulted in the formation of inclusion bodies. To address this challenge, we implemented split intein-mediated *trans*-splicing to deliver protein cargos onto the surface of ferritin nanoparticles. Using this approach, both rhizavidin and CBLB502 were efficiently coupled to ferritin nanoparticles. Furthermore, we demonstrated the concurrent conjugation of antigens and adjuvant molecules within a single ferritin nanoparticle, resulting in more potent antibody responses than mixtures of nanovaccine and adjuvant [119]. Building upon this success, we genetically fused CBLB502 to the C-terminus of ferritin and utilized split intein-mediated *trans*-splicing to load antigens onto its N-terminus. Trimeric spikes of SARS-CoV-2 were then displayed onto ferritin nanoparticles with CBLB502 at their C-terminus, creating an antigen/adjuvant co-delivery nanoparticle capable of inducing robust mucosal and systemic immune responses [120]. Furthermore, SpyCatcher/SpyTag technology was also applied to modify ferritin nanoparticles, displaying the gE of varicella–zoster virus (VZV) at its N-terminus. This nanovaccine elicited 3.2-fold higher neutralizing antibody titers compared to gE protein alone in BALB/c and C57BL/6 mice [121].

### 3.8. Lumazine Synthase (LS) Nanoparticle as a Vaccine Nanocarrier

The LS of Aquafex aeolicus (AaLS) spontaneously self-assembles into exceptionally thermostable (melting temperature 119.9 °C) 60-mer (T = 1) icosahedral nanoparticles [122]. AaLS recognizes a specific 12-amino acid motif at the C-terminus of homologous riboflavin synthase, enabling it to encapsulate guest proteins carrying this motif and form a nanocompartment. This biomimetic property has been widely applied in medical and biological fields [122]. AaLS self-assembling nanoparticles allow for the N-terminal insertion of the SARS-CoV-2 RBD, eliciting significantly higher neutralizing antibody titers than RBD monomers in mice [107]. Furthermore, AaLS nanoparticles accommodate the C-terminal genetic fusion of the ectodomain of HIV gp120, triggering sustained B cell responses [108]. To develop a universal platform for antigen delivery, SpyTag has been strategically placed at the N-terminus of AaLS, allowing for the covalent conjugation of trimers of respiratory syncytial virus (RSV) prefusion (preF) or trimers of SARS-CoV-2 spike fused with SpyCatcher [123]. Additionally, SpyCatcher has been fused to the N-terminus of AaLS for covalent coupling of monkeypox antigens L1, A29, or A33, expressed in *E. coli* with SpyTag at their N-termini. All three nanoparticles successfully induced high titers of antibodies [124]. Beyond its natural self-assembly into 16 nm spherical nanoparticles, AaLS has been engineered to form rod-shaped or spherical nanoparticles of varying sizes. By strategically relocating amino acids 120–156 from the C-terminus of AaLS to the N-terminus of amino acids 1–119, with the insertion of L8H4, L12H6, or L16H6 linkers, researchers have successfully controlled the assembly of quaternary structures [125].

### 3.9. Encapsulin Nanoparticle as a Vaccine Nanocarrier

Encapsulins, a group of proteins found in prokaryotic organisms, are capable of recognizing and encapsulating guest proteins carrying specific cargo-loading peptide (CLP) signals. These proteins spontaneously assemble into nanoparticles ranging in size from 25 nm to 42 nm [126]. When a 6xHis tag was fused to the C-terminus of *Thermotoga maritima* (TM) encapsulin, nanoparticle formation occurred; however, the target product did not exhibit specific binding to nickel resin, suggesting that the 6xHis tag extended into the interior cavity of the nanoparticle. Instead, inserting the 6xHis tag between amino acids 42 and 43 within an exposed loop had no adverse effect on nanoparticle stability and enabled purification via nickel chromatography. This modified nanoparticle also allowed for the insertion of the SP94 peptide, which specifically targets hepatocellular carcinoma (HCC) cells, between amino acids 138 and 139 without disrupting nanoparticle assembly [109]. Encapsulin has been further developed as a nanocarrier platform by genetically fusing SpyCatcher to its C-terminus, enabling the covalent coupling of the RBD of the SARS-CoV-2 BA.5 strain containing a SpyTag. This nanoparticle elicits high titers of neutralizing antibodies with a broad spectrum in mice, making it a promising candidate for next-generation COVID-19 vaccines [127]. Additionally, we engineered encapsulin as a modular nanocarrier by introducing split intein^N^ to its C-terminus, facilitating the covalent conjugation of antigens carrying a complementary split intein^C^ at their N-terminus. The direct fusion of split intein^N^ to encapsulin (encapsulin-int^N^) resulted in the formation of inclusion bodies. Introducing a single gb1 or halo tag to the C-terminus of encapsulin-int^N^ slightly enhanced recombinant protein solubility, while incorporating both tags significantly improved solubility, resulting in a highly soluble recombinant protein. GFP protein and three other proteins were successfully coupled to the encapsulin nanocarrier, inducing notably higher antigen-specific antibody responses compared to the proteins alone [128].

### 3.10. mI3 Nanoparticle as a Vaccine Nanocarrier

In 2016, a computational design method was employed to optimize the nanoparticle formed by 60 units of KDPG, resulting in a mutant named I3-01 with five substitutions (E26K, E33L, K61M, D187V, and R190A), which stabilized the interface between trimers. This newly designed nanoparticle exhibited enhanced stability at 80 °C or in 6.7 M guanidine hydrochloride (GuHCl) and allowed for the genetic fusion of GFP to its N-terminus, C-terminus, or both [110]. When SpyCatcher was inserted into the N-terminus of I3-01, two protein bands were observed, likely due to improper disulfide bond formation. This heterogeneity was eliminated by mutating two exposed cysteines on the nanoparticle surface to alanines (C76A and C100A), resulting in SpyCatcher-mI3. A 6xHis tag was further added to the C-terminus of SpyCatcher-mI3 for purification purposes. This engineered nanoparticle facilitates the display of desired antigens and shows significant potential in eliciting robust IgG responses [27]. In another study, the RBD of SARS-CoV-2 was displayed on the surface of mI3 nanoparticles using SpyCatcher/SpyTag technology, resulting in a nanovaccine that induced remarkably elevated neutralizing antibody titers compared to RBD alone [129]. In addition to mI3, SpyCatcher/SpyTag technology has also been applied to AP205 and ferritin nanoparticles. gE of VZV has been covalently coupled to all three nanoparticle platforms, with each eliciting comparable and robust IgG responses. Interestingly, gE delivered by mI3 nanoparticles triggered significantly stronger IFN-γ responses compared to ferritin or AP205 nanoparticles in mice. The linker inserted between gE and SpyTag has an effect on proteins’ yield and their immunogenicity. The linker inserted between gE and SpyTag was found to influence protein yield and immunogenicity. A recombinant protein with a rigid EAAAK linker showed the highest expression levels and elicited stronger IgG and T-cell responses compared to proteins with a flexible GGGGS linker or no linker [130].

### 3.11. Development of Norovirus VP1-Derived Nanoparticles as Vaccine Nanocarriers

The full-length norovirus VP1 protein has been developed as a modular nanocarrier platform by introducing SpyTag to its C-terminus, enabling covalent coupling of the extracellular domain of influenza virus matrix protein M2 (M2e) or the hemagglutinin stem domain HA2 containing SpyCatcher at their N-terminus. While HA2 delivered by full-length VP1 nanoparticles elicited high IgG antigen-specific titers, this nanovaccine failed to induce neutralizing antibodies. Similarly, M2e displayed by VP1 nanoparticles did not generate antigen-specific antibodies in mice [104]. These findings suggest that the antigens delivered by full-length VP1 might be improperly oriented or displayed, potentially due to their distribution within the grooves of the nanoparticle rather than on its surface.

Beyond the full-length norovirus VP1, the S domain and P domain of VP1 have also been explored as nanocarrier platforms [131]. In 2018, the S domain of GII.4 VP1 was expressed in *E. coli* with a GST tag at its N-terminus, and nanoparticles were detected by gel filtration and Transmission Electron Microscope (TEM) analysis. However, an extra degraded protein band was observed, which was eliminated by introducing the R69N mutation. Inserting a 6xHis tag at the C-terminus of the S nanoparticle resulted in a 60-mer nanoparticle (S_60_-6xHis), confirmed by cryo-EM analysis. However, attempts to insert the human rotavirus VP8 antigen at the C-terminus of the S domain compromised nanoparticle assembly. This challenge was addressed by introducing three cysteines at residues V57C, Q58C, and S136C to form inter-molecular disulfide bonds, significantly improving nanoparticle formation efficiency. The resulting S_60_-VP8 nanoparticle induced remarkably higher antigen-specific and neutralizing antibody titers compared to VP8 proteins in mice [28]. Furthermore, the S_60_ nanoparticle platform has been successfully employed to display various antigens, including the head domain of H7N1 HA (HA1) [132], VP8 antigen of the murine rotavirus EDIM strain [133], and the TSR antigen of the circumsporozoite surface protein (CSP) of the malaria parasite *Plasmodium falciparum* [134], all of which elicited significantly stronger immune responses than protein-based antigens. However, challenges persist with the S_60_ platform. When the RBD of SARS-CoV-2 was fused to the C-terminus of S_60_, recombinant protein in *E. coli* formed inclusion bodies, which could be solubilized using 4 M urea. Refolding by dialysis resulted in heterogeneous nanoparticles with different sizes. Expression of S_60_-RBD in CHO cells also produced heterogeneous nanoparticles with sizes of 25 nm (T = 1) and 35 nm (T = 3) [135]. Similarly, delivery of the VP4e antigen of human rotavirus, while eliciting significantly higher antibody responses than proteins, resulted in heterogeneous nanoparticles with different sizes (20 nm, 28 nm, and 34 nm) when expressed in *E. coli* [136]. Furthermore, inserting a fused NSP4-VP8 antigen of human rotavirus into the C-terminus of S resulted in the aggregation of large complexes rather than spherical nanoparticles [137]. In summary, the S_60_ platform demonstrates promise for delivering various antigens, but certain antigens can compromise nanoparticle assembly. Further research is needed to improve the stability of the S_60_ nanoparticle and address the adverse effects of certain antigens on its assembly.

In 2005, the P domain of norovirus was engineered as a 24-mer nanoparticle by introducing an oligopeptide containing one or more cysteines at its C-terminus. Compared to the P dimer, the P_24_ nanoparticle exhibited a >700-fold increase in histo-blood group antigen (HBGA) binding sensitivity [138]. The P_24_ nanoparticle tolerated the insertion of a 6xHis tag into exposed loop sites between amino acids 372 and 374, enabling the incorporation of the human rotavirus VP8 antigen between amino acids 365–375 [139]. Furthermore, VP8 antigens of P[8], P[4], and P[6] rotaviruses were displayed by P24 nanoparticles to generate a trivalent vaccine that induced high antigen-specific and neutralizing antibody titers [140].

## 4. Conclusions

Nanotechnology holds significant promise for developing next-generation vaccines by mimicking the biomimetic properties of natural pathogens. Nanoparticles, designed to resemble pathogen size, geometry, and antigen density, enhance antigen delivery to lymph nodes, uptake by APCs, and presentation to B cells, ultimately leading to robust and long-lasting immune responses. The *E. coli* expression system is highly favored for nanoparticle production due to its high yield, short production cycle, and low cost. However, challenges remain, including incorrect disulfide bond formation, the absence of post-translational modifications, and improper nanoparticle assembly. To obtain properly assembled VLPs from *E. coli* expression system, researchers have implemented various strategies: deletion of unnecessary residues, utilization of molecular chaperone, and disassembly and reassembly treatment. In addition to 9-valent HPV vaccines, higher-valent HPV vaccines have been developed, such as 15-valent HPV, which pose great stress on manufacturing. Constructing hybrid nanoparticles or swapping similar loops to generate chimeric nanoparticles provides an alternative approach to producing higher-valent HPV VLPs in an easy manner.

To optimize the design of nanoparticle-based vaccines, several critical considerations are crucial. First, different nanocarriers and linker types can influence the immunogenicity of delivered antigens, necessitating optimization. Second, the insertion site of loaded cargo can affect nanoparticle assembly, and improper antigen display can lead to insufficient immune responses. Third, the stability of the nanocarrier can be improved by introducing artificial disulfide bridges or substitutions of residues between the interface. Beyond genetic fusion, chemoenzymatic methods offer alternative strategies for displaying antigens on nanoparticle surfaces, providing more flexibility in antigen design and allowing for the integration of diverse antigens (Figure 5). Vaccines like LYB001 and LYB004, which utilize SpyCatcher/SpyTag technology for antigen display, demonstrate the successful translation of these advanced strategies into clinical trials. This highlights the potential of chemoenzymatic methods to create novel nanoparticle vaccines with improved efficacy and safety. These ongoing advancements in nanotechnology and chemoenzymatic methods, coupled with the advantages of *E. coli*-based production, hold significant promise for developing highly effective and affordable nanovaccines to address a wide range of infectious diseases.

**Figure 5 vaccines-12-01287-f005:**
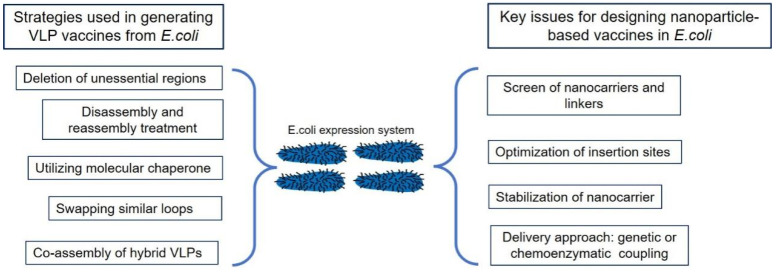
Advanced nanotechnologies used in *E. coli*-derived vaccine development.

## Figures and Tables

**Figure 1 vaccines-12-01287-f001:**
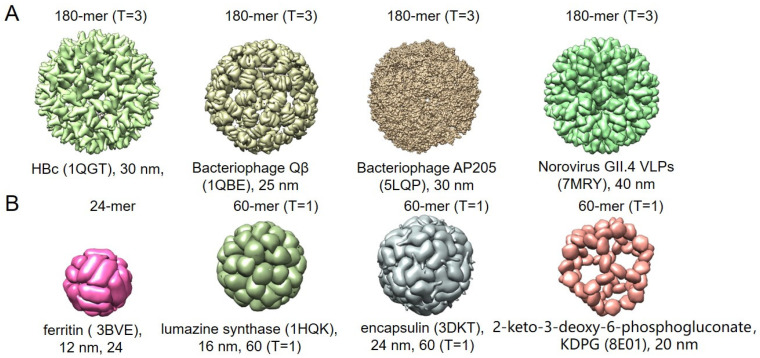
Commonly used nanocarriers to display antigens. (**A**) 3D structure of VLPs. (**B**) 3D structure of protein-based nanoparticles. T represents the triangulation of nanoparticles, and mer is the abbreviation of protomers.

**Figure 2 vaccines-12-01287-f002:**
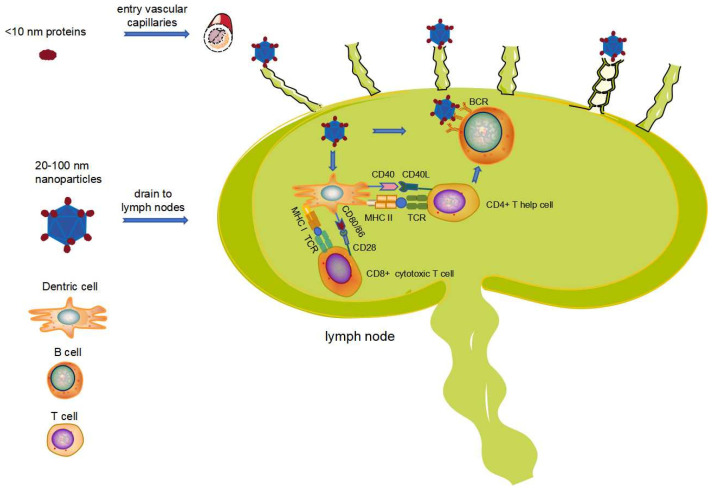
Nanoparticles offer several key advantages over conventional vaccines, leading to more potent and targeted immune responses. Unlike smaller proteins or particles (<10 nm) that are quickly cleared from the bloodstream, nanoparticles (20–100 nm) efficiently target draining lymph nodes, the primary site of immune response initiation. Furthermore, nanoparticles are preferentially engulfed by antigen-presenting cells (APCs), particularly dendritic cells (DCs), maximizing antigen presentation and immune activation.

**Figure 3 vaccines-12-01287-f003:**
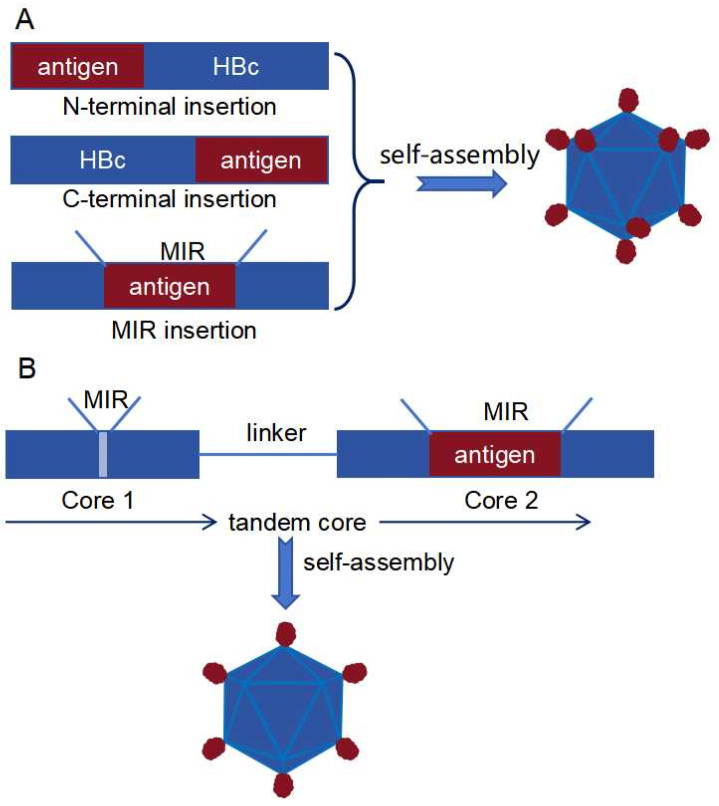
Display of antigens using the genetic fusion approach. (**A**) Antigens can be inserted at the N-terminus, C-terminus or MIR site of HBc. (**B**) Two consecutive HBc are designed as nanocarriers with the insertion of a single antigen at the MIR site to reduce steric hindrance.

**Figure 4 vaccines-12-01287-f004:**
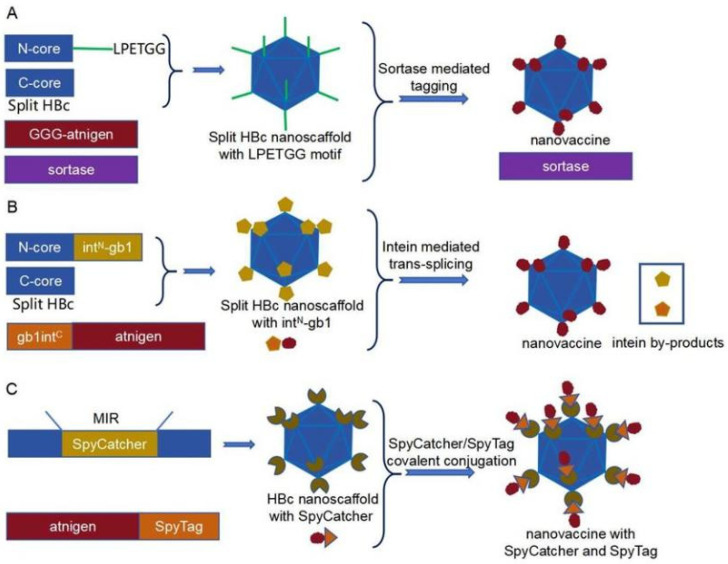
Three chemoenzymatic methods to loading antigens onto HBc nanocarrier. (**A**) Soratase-mediated site-specific tagging is introduced into split HBc nanoscaffold. (**B**) Split intein-mediated *trans*-splicing is utilized to conjugate antigens onto split HBc nanoscaffold. (**C**) SpyCatcher/SpyTag mediated covalent conjugation is applied to couple antigens onto HBc nanoscaffold.

**Table 1 vaccines-12-01287-t001:** *E. coli*-derived VLP or nanoparticle vaccines licensed or in clinical trials.

Vaccines	VLP/Nanoparticle Platforms	Vaccine Antigens	Clinical Trial/ Approved	References/Clinical Trial Identifiers
HEV Hecolin^®^	Hepatitis E virus (HEV)	p239 of HEV capsid protein	Licensed	[17]
HPV Cecolin^®^ 2	Human papillomavirus (HPV)	HPV16/18 L1 major capsid protein	Licensed	[18]
HeberNasvac (therapeutic vaccines HBV ABX203) ^a^	Hepatitis B surface antigen (HBsAg)/hepatitis B core antigen (HBcAg)	HBsAg/HBcAg	Licensed	[19]
HPV Cecolin^®^ 9	HPV	HPV6/11/16/18/31/33/45/52/58 L1 major capsid protein	Phase 3	NCT05056402, NCT04782895
LYB001 ^b^	I3-01	SARS-CoV-2 RBD	Phase 3	NCT05664932, NCT05683600
LYB004 ^b^	I3-01	Varicella–zoster virus (VZV) gE	Phase 1	NCT06335849
gH1-Qbeta ^c^	Qβ	Globular head domain (gH1) of influenza A hemagglutinin (HA)	Phase I	[20]
ACAM-FLU-A ^d^	HBcAg	Influenza A M2e	Phase 1	NCT00819013
ICC-1132 (MalariVax) ^d^	HBcAg	The circumsporozoite protein of *Plasmodium falciparum*	Phase I	NCT00587249

^a^ HeberNasvac is composed of HBsAg and HBcAg, which are produced by yeast and *E. coli*, respectively. ^b^ Antigens are displayed on the I3-01 nanoparticle using SpyCatcher/SpyTag technology. ^c^ gH1 is delivered onto Qβ VLP by chemical conjugation. ^d^ Antigens are genetically fused to HBcAg and presented on the surface of VLP.

## Data Availability

The data are available from the corresponding author upon reasonable request.

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
