# Peer review of "Engineering *Escherichia coli*-Derived Nanoparticles for Vaccine Development"

_vaccines, 2024, doi:10.3390/vaccines12111287_

Round 1
Reviewer 1 Report
Comments and Suggestions for Authors
This review article titled “Engineering E.coli-derived nanoparticle for vaccine development” is interesting and well written. Below are some comments to be addressed.
It would be interesting to the researchers if the authors could include tabulate a summary of all the ongoing clinical trial using this approach of Engineering E.coli-derived nanoparticle for vaccines.
Also, it would help if the authors could tabulate what vaccines are already approved using the E-coli derived technology and their indications.
Fig.5 shows the advanced nanotechnologies used in E.coli-derived vaccine development. Did any of these advanced technologies translate from for bench to bedside yet? Could the authors comment on future developments and anticipated timeline on potential health authority approvals?
Author Response
Response to reviewer 1:
Comments and Suggestions for Authors:
This review article titled “Engineering E.coli-derived nanoparticle for vaccine development” is interesting and well written. Below are some comments to be addressed.
Responses to general comments and suggestions: We appreciate your positive assessment of our work and your valuable suggestions for strengthening the significance of our manuscript. We have revised our manuscript to address the comments needed.
Comment 1: It would be interesting to the researchers if the authors could include tabulate a summary of all the ongoing clinical trial using this approach of Engineering E.coli-derived nanoparticle for vaccines.
Response 1: This suggestion is helpful to demonstrate the significance of engineering E.coli-derived nanoparticle for vaccines by listing ongoing clinical trial. We have added a new table (‘Table 1’) summarizing licensed and ongoing clinical trials of E. coli-derived nanoparticle vaccines, highlighting the translational potential of this approach.
Comment 2: Also, it would help if the authors could tabulate what vaccines are already approved using the E-coli derived technology and their indications.
Response 2: This advice is helpful and vaccines approved using E-coli derived technology are listed in “Table 1”. We have included a new sentence in the third paragraph of the ‘Introduction’ section, listing the licensed and ongoing clinical trials of E. coli-derived nanoparticle vaccines, to emphasize the platform’s potential for addressing global health challenges.
Comment 3: Fig.5 shows the advanced nanotechnologies used in E.coli-derived vaccine development. Did any of these advanced technologies translate from for bench to bedside yet? Could the authors comment on future developments and anticipated timeline on potential health authority approvals?
Response 3: This suggestion is useful to elucidate the translational value of the advanced technologies to clinical trials. We have updated the second paragraph of the ‘Conclusion’ section to highlight the successful translation of chemoenzymatic methods into clinical trials, emphasizing the potential of SpyCatcher/SpyTag technology for creating novel nanoparticle vaccines with improved efficacy and safety.
Reviewer 2 Report
Comments and Suggestions for Authors
Tang et al present an exhaustive review on the production of vaccine nanoparticles in Escherichia coli.
They cover all the most popular nanoparticle scaffolds and the different approaches (genetic and chemical) that can be employed for antigen/epitope conjugation and display.
Some comments and suggestions for revision are listed below.
1. Except for the VP1 VLP-based norovirus vaccine HIL-214, no mention is made about possible ongoing clinical studies on all the other nanoparticle vaccine candidates described by the authors. None of them is actually been subjected to clinical evaluation? If so, this should be clearly stated somewhere in the text.
2. The numbers in Figure 2 legend (1, 2 ,3) are not connected to any specific panel numbering in the figure and points #2 and #3 are actually identical. Please check and correct as appropriate.
3. At page 5 it is not clear what the authors mean by “paired L2 monomers”. It would be clearer to just say that there is an estimated number of 12 L2 molecules/HPV capsid and use a more updated reference to support this information.
4. The heading of the “Antigens” column in Table 1 should be changed and made consistent with the general title of this Table (“Genetic fusion of desired molecules…”). In fact, some of the mentioned fusion partners, such as the 6xHis tag, SpyTag and SpyCatcher, but also GFP, cannot be viewed as real antigens.
5. The “formed” adjective in the headings of sections 3.7-3.10 doesn’t make sense to me and it should be deleted; “XXXX nanoparticles as a vaccine nanocarrier” reads much better.
6. The “LS” acronym, which stands for Lumazine Synthase, reported in brackets in the heading of section 3.9 is probably a typo and should be deleted or corrected.
7. The terms “nanovirus” and “nanacarriers” in the heading of section 3.11 are both likely mistakes and should be fixed.
Author Response
Response to reviewer 2:
Comments and Suggestions for Authors
Tang et al present an exhaustive review on the production of vaccine nanoparticles in Escherichia coli.
They cover all the most popular nanoparticle scaffolds and the different approaches (genetic and chemical) that can be employed for antigen/epitope conjugation and display.
Some comments and suggestions for revision are listed below.
Response to general comments and suggestions:
We appreciate your positive assessment of our manuscript and your helpful suggestions for improving its clarity and accuracy.
Comment 1. Except for the VP1 VLP-based norovirus vaccine HIL-214, no mention is made about possible ongoing clinical studies on all the other nanoparticle vaccine candidates described by the authors. None of them is actually been subjected to clinical evaluation? If so, this should be clearly stated somewhere in the text.
Response 1: We have added a new table (“Table 1”) summarizing licensed and ongoing clinical trials of E. coli-derived nanoparticle vaccines to address your point about the translational potential of this technology.
Comment 2. The numbers in Figure 2 legend (1, 2 ,3) are not connected to any specific panel numbering in the figure and points #2 and #3 are actually identical. Please check and correct as appropriate.
Response 2: We have revised the numbering in Figure 2 legend to ensure consistency with the figure panels and have corrected the error where points #2 and #3 were identical.
Comment 3: At page 5 it is not clear what the authors mean by “paired L2 monomers”. It would be clearer to just say that there is an estimated number of 12 L2 molecules/HPV capsid and use a more updated reference to support this information.
Response 3: We have clarified the description of L2 molecules within the HPV capsid, stating that the number of L2 is controversial for each capsid and providing a more recent reference to support this information.
Comment 4: The heading of the “Antigens” column in Table 1 should be changed and made consistent with the general title of this Table (“Genetic fusion of desired molecules…”). In fact, some of the mentioned fusion partners, such as the 6xHis tag, SpyTag and SpyCatcher, but also GFP, cannot be viewed as real antigens.
Response 4: We have changed the heading of the “Antigens” column in Table 1 to “Molecules” to be consistent with the general title of the table, “Genetic fusion of desired molecules…” We have also revised the previous Table 1 to become Table 2.
Comment 5: The “formed” adjective in the headings of sections 3.7-3.10 doesn’t make sense to me and it should be deleted; “XXXX nanoparticles as a vaccine nanocarrier” reads much better.
Response 5: We have removed the word ‘formed’ from the headings of sections 3.7-3.10, resulting in a more concise and clear title like “XXXX nanoparticles as a vaccine nanocarrier”.
Comment 6: The “LS” acronym, which stands for Lumazine Synthase, reported in brackets in the heading of section 3.9 is probably a typo and should be deleted or corrected.
Response 6: We have corrected the “LS” acronym to “Lumazine Synthase” in the heading of section 3.9.
Comment 7: The terms “nanovirus” and “nanacarriers” in the heading of section 3.11 are both likely mistakes and should be fixed.
Response 7: We have corrected the terms “nanovirus” and “nanocarriers” in the heading of section 3.11 to the appropriate terminology.
Reviewer 3 Report
Comments and Suggestions for Authors
Title: Engineering E. coli-derived nanoparticle for vaccine development
This manuscript is well-written by the authors. I do believe that if they can improve the manuscripts following all comments. It might have a chance to publish in the journal.
Comments
1. Topic: Please re-write the topic. Please use the full name of E. coli.
2. It would be better if the authors add the line number in the manuscript. It is easy to read and comment by reviewer.
3. Please write the scientific name in italic.
4. Please add the objective of the review in the abstract.
5. Keywords: It would be better if the authors use simple words as the keyword instead of the phrase.
6. Introduction: The first paragraph is mentioned about SARS-CoV-2. What is the link or relationship between the E-coli-vaccine and SARS-CoV-2? Please describe.
7. Please add the objective of the review in the end of introduction.
8. Figure 1: It would be better if the authors add the figure caption in the same page of the figure.
9. “3.1. Development of E.coli-derived HEV Vaccines”: The genome of HEV is a single-stranded positive-sense RNA molecule.
10. “3. Development of E.coli-derived nanovaccines”: I suggest the authors to prepare 1 Table for this section about the information of each E.coli-derived nanovaccines. It will be easy to understand.
11. Please modify or re-write conclusion. It should be summarized on the key finding. Please don’t repeat the results and discussion. Please summarize by your own information. No need to add the citation.
12. The references of 2020-2024 are suggested to be cited. Please remove some old references and unnecessary references.
Author Response
Responses to reviewer 3
Comments and Suggestions for Authors
Title: Engineering E. coli-derived nanoparticle for vaccine development
This manuscript is well-written by the authors. I do believe that if they can improve the manuscripts following all comments. It might have a chance to publish in the journal.
Response to comments and suggestions: We appreciate your positive assessment of our manuscript and your valuable suggestions for improving its clarity and accuracy.
Comment 1. Topic: Please re-write the topic. Please use the full name of E. coli.
Response 1: We have revised the title to read “Engineering Escherichia coli-derived nanoparticles for vaccine development” to reflect the full name of the bacterium.
Comment 2: It would be better if the authors add the line number in the manuscript. It is easy to read and comment by reviewer.
Response 2: We understand the value of line numbers for reviewer communication. However, as per the editor’s instructions, we have maintained the current typesetting of the manuscript for publication.
Comment 3: Please write the scientific name in italic.
Response 3: We have carefully checked and corrected the scientific names to ensure they are italicized throughout the manuscript.
Comments 4: Please add the objective of the review in the abstract.
Response 4: We have corrected the last sentence to the abstract, which clearly states the objective of this review: “These bioengineering approaches, in combination with advanced nanocarrier design, hold significant potential for developing highly effective and affordable E. coli-derived nanovaccines, paving the way for improved protection against a wide range of infectious disease.” .
Comments 5: Keywords: It would be better if the authors use simple words as the keyword instead of the phrase.
Response 5: We have revised the keywords, replacing any phrases with individual words, to improve searchability and clarity.
Comments 6: Introduction: The first paragraph is mentioned about SARS-CoV-2. What is the link or relationship between the E-coli-vaccine and SARS-CoV-2? Please describe.
Response 6: We have added a sentence to the third paragraph of the “Introduction” section to highlight the relevance of E. coli-derived vaccines to SARS-CoV-2. The sentence explains that LYB001, an E. coli-derived VLP vaccine targeting the receptor-binding domain (RBD) of SARS-CoV-2, has entered clinical phase III trials, demonstrating the platform’s potential for addressing emerging infectious diseases.
Comments 7: Please add the objective of the review in the end of introduction.
Response 7: We have included a new sentence at the end of the “Introduction” section, outlining the scope of the review: “The subsequent sections will delve into the lessons learned and engineering technologies employed to address these challenges and optimize the rational display of antigens onto nanocarriers, maximizing vaccine efficacy.”
Comments 8: Figure 1: It would be better if the authors add the figure caption in the same page of the figure.
Response 8: We have moved the figure caption to the same page as Figure 1 to enhance readability.
Comments 9: “3.1. Development of E.coli-derived HEV Vaccines”: The genome of HEV is a single-stranded positive-sense RNA molecule.
Response 9: We have corrected the description of HEV, accurately stating that its genome is a single-stranded positive-sense RNA molecule.
Comments 10: “3. Development of E.coli-derived nanovaccines”: I suggest the authors to prepare 1 Table for this section about the information of each E.coli-derived nanovaccines. It will be easy to understand.
Response 10: We have created a new Table 1, which summarizes licensed and in clinical trials E. coli-derived nanoparticle vaccines. We have also compiled information on the most commonly used nanocarriers in Table 2.
Comments 11: Please modify or re-write conclusion. It should be summarized on the key finding. Please don’t repeat the results and discussion. Please summarize by your own information. No need to add the citation.
Response 11: We have revised the “Conclusion” section to provide a concise summary of key findings and future directions for E. coli-derived nanoparticle vaccines, drawing on our understanding of the field’s achievements and trends.
Comments 12. The references of 2020-2024 are suggested to be cited. Please remove some old references and unnecessary references.
Response 12: We have updated several references to include more recent publications from 2020-2024. However, we have retained some older references that provide valuable insights into the historical development and technological advancements in E. coli-derived nanoparticle vaccines.
Reviewer 4 Report
Comments and Suggestions for Authors
This article provides an review of molecular and protein engineering strategies for the development of nanovaccines based on recombinant pethides produced by Escherichia coli. The paper highlights recent trends and successes in this field. The manuscript discusses advances and prospects for the development of new vaccines using virus-like particles and ferritin nanoparticles as examples The review will certainly be of interest to researchers seeking to optimize the production of vaccine nanoparticles and to non-specialists in the field
The paper is recommended for publication after some corrections. The lack of line numbering makes it difficult to address comments.
1.The term “nanoparticle vaccines” should be defined in the introduction
2. The abbreviation “VLP” should be decoded
3.The review should avoid specifying concentrations of solutions, composition of buffers for isolation.
4. What is meant by T=3, T=1? If you give these values, the reader should know what they mean.
5. The legend to Figure 1 should be under the figure, not on another page
6. Please explain in more detail what is the “reverse targeted” cell infiltration phenomenon?
7. The text incorrectly says that “host tRNAs were removed following TEV digestion”.
8. It is necessary to correct Table 1
9. Correct the font of Helicobacter pylori
Author Response
Responses to reviewer 4
Comments and Suggestions for Authors
This article provides a review of molecular and protein engineering strategies for the development of nanovaccines based on recombinant pethides produced by Escherichia coli. The paper highlights recent trends and successes in this field. The manuscript discusses advances and prospects for the development of new vaccines using virus-like particles and ferritin nanoparticles as examples. The review will certainly be of interest to researchers seeking to optimize the production of vaccine nanoparticles and to non-specialists in the field.
The paper is recommended for publication after some corrections. The lack of line numbering makes it difficult to address comments.
Response to general comments and suggestions: We appreciate your positive assessment of our manuscript and your helpful suggestions for improving its clarity and accuracy. We understand the value of line numbers for reviewer communication. However, as per the editor’s instructions, we have maintained the current typesetting of the manuscript for publication.
Comment 1: The term “nanoparticle vaccines” should be defined in the introduction
Response 1: We have added a definition of “nanoparticle vaccines” to the first paragraph of the ‘Introduction’ section, explaining that they mimic the size, geometry, and multivalent antigen presentation of native pathogens, enabling a robust immune response without compromising safety.
Comment 2: The abbreviation “VLP” should be decoded
Response 2: We have included the full name of ‘VLP’ (virus-like particle) for clarity..
Comment 3: The review should avoid specifying concentrations of solutions, composition of buffers for isolation.
Response 3: We have removed detailed information about specific concentrations of solutions and buffer compositions from the first paragraph of section 3.2, focusing on the broader principles of HPV VLP disassembly and reassembly.
Comment 4: What is meant by T=3, T=1? If you give these values, the reader should know what they mean.
Responses 4: We have added a definition of the T number to the Figure 1 legend: “T represents the triangulation number of nanoparticles”, indicating the arrangement of protein subunits in the capsid. “mer refers to protomers”, the basic building blocks of the capsid.”
Comment 5: The legend to Figure 1 should be under the figure, not on another page
Response 5: We have moved the figure caption to the same page as Figure 1 to enhance readability.
Comment 6: Please explain in more detail what is the “reverse targeted” cell infiltration phenomenon?
Response 6: We have expanded the explanation of the “reverse targeted” cell infiltration phenomenon in the first paragraph of Section 2. We now clarify that dendritic cells (DCs) engulf the antigen-adjuvant complex, mature, and migrate back to the lymph nodes through a process called chemokine homing. This process effectively delivers the antigen to the lymph nodes, where it can be presented to T cells and B cells.
Comment 7: The text incorrectly says that “host tRNAs were removed following TEV digestion”.
Response 7: We have corrected the text in Section 3.3 to accurately state that “tRID was cleaved from the recombinant protein following TEV digestion, along with host tRNA.”
Comment 8: It is necessary to correct Table 1
Response 8: We have created a new Table 1, which summarizes licensed and in clinical trials E. coli-derived nanoparticle vaccines. We have also corrected the errors in previous “Table 1” which is named a “Table 2” new.
Comment 9: Correct the font of Helicobacter pylori
Response 9: We have corrected the font of Helicobacter pylori to ensure proper italicization of the scientific name.